# A follow-up study (2007–2018) on American Tegumentary Leishmaniasis in the municipality of Caratinga, Minas Gerais State, Brazil: Spatial analyses and sand fly collection

**Rafael L. Neves**[1]☉, **Diogo T. Cardoso**[2]☉, **Felipe D. Rêgo**[1], **Célia M. F. Gontijo**[1], **David S. Barbosa**[2], **Rodrigo P. Soares**[1]*

1 René Rachou Institute, Oswaldo Cruz Foundation, Belo Horizonte, Minas Gerais, Brazil, 2 Parasitology Department, Institute of Biological Sciences, Federal University of Minas Gerais, Belo Horizonte, Minas Gerais, Brazil

☉ These authors contributed equally to this work.
* rodrigosoares28@hotmail.com

## Abstract

### Background

The municipality of Caratinga is an important endemic area for American Tegumentary Leishmaniasis (ATL) and no epidemiological studies were performed during the past two decades. Here, we analyzed the epidemiological situation and the geographical distribution of ATL cases in the municipality of Caratinga from 2007 to 2018 using geographic information systems (GIS). Also, we evaluated the impact of several demographic parameters in ATL distribution and the sand flies incriminated in its transmission.

### Methods

All demographic information (gender, age, educational level, clinical form, diagnostic criteria and case evolution) used in this study was retrieved from the public health archives and confirmed in the State Health Services databases. All cases were analyzed using GIS software based on ATL distribution. Also, non-systematic sand fly collections and molecular detection of *Leishmania* were performed in the hotspots.

### Results and conclusions

During the period, ATL cases continued and increased especially in the past years (2016–2018). Hotspots included urban Caratinga areas and the districts of Patrocínio de Caratinga and Sapucaia. The species *Nyssomyia whitmani*, *Nyssomyia intermedia*, *Migonemyia migonei* and *Evandromyia cortelezzii* complex were captured. However, ITS1-PCR did not detect *Leishmania* DNA in those insects. Based on our analyses, urbanization of ATL in Caratinga has occurred in the past years. Due to the increase in the number of cases and vectors presence, it is recommended that health authorities focus on control measures in the most affected areas (Patrocínio of Caratinga and Sapucaia districts and urban Caratinga).

**Data Availability Statement:** All relevant data are within the manuscript and its Supporting Information files.

**Funding:** RPS and CMFG are research fellows of Conselho Nacional de Desenvolvimento Científico e Tecnológico (CNPq) (302972/2019-6; 305430/2017-3) (www.cnpq.br) RLN was funded by Cooordenação de Aperfeiçoamento do Ensino Superior (CAPES) (8881.309862/2018-01)(https://www.gov.br/capes/pt-br). This study was partially supported by the Coordination for the Improvement of Higher Education Personnel Vice-Presidência de Educação, (Coordenação de Aperfeiçoamento de Pessoal de Nível Superior - CAPES) - Finance Code 001. The funders had no role in study design, data collection and analysis, decision to publish, or preparation of the manuscript.

**Competing interests:** The authors have declared that no competing interests exist.

## Author summary

Leishmaniasis is an important health problem in Latin American countries and worldwide. In some places, notification is limited and underreported hindering correct assessment of existing data. In Brazil, ATL is mainly caused by *Leishmania* (*Viannia*) *braziliensis*. It is transmitted by sand fly vectors *Nyssomyia whitmani* and *Nyssomyia intermedia*. ATL comprises different clinical manifestations including cutaneous leishmaniasis, muco-cutaneous leishmaniasis and atypical leishmaniasis. The objective is to present the spatial distribution of cases of ATL notified by the Public Health System of Caratinga between the period of 2007 and 2018. This city is one of the most important endemic areas in the State of Minas Gerais and compose the panel of knowledge on epidemiological aspects of cutaneous leishmaniasis caused by *L. braziliensis*. This information will be important for developing strategies for ATL control in the affected areas. There is strong evidence that supports ATL urbanization in the city of Caratinga and increase of cases in two rural districts (Sapucaia and Patrocínio de Caratinga).

## Introduction

Caratinga is a city located in the Rio Doce Valley in the eastern part of Minas Gerais State, Brazil. It is an urbanized city surrounded by a vast rural area of coffee crops. For this reason, a high number of ATL cases are reported mainly in rural workers occupationally exposed. Outbreaks of ATL have been occurred since the 1960s and this phenomenon attracted pioneer researchers to this endemic area. They were conducted by Dr. Mayrink´s group and most of the knowledge regarding ATL management in Brazil was originated from these studies [1].

Historically, the presence of his group together with the local Health authorities was responsible for creating a Reference Center on Leishmaniasis in the town. This facility enabled to diagnose, treat, and follow ATL patients for several decades [2]. Besides, several eco-epidemiological studies on sand fly vectors, reservoirs and immunotherapy trials were also developed in the area helping to control the disease [3,4]. Those measures included early diagnosis, patient treatment and spraying houses with insecticides and successfully controlled ATL in some areas [5].

Regarding patient management, the reference center of Caratinga was very important for standardization of Montenegro´s skin test that is currently obsolete [4,6,7]. This test was successfully employed in field trials for testing a so-called vaccine against the disease. In fact, the vaccine consisted of a mixture of *Leishmania* parasite strains/species used as immunotherapy, a procedure that is no longer available. Most of the patients were still positive after these trials suggesting long-lasting immune response. In these patients, no ATL episodes were reported in the following years after the research [8]. Later, those trials were also evaluated in other endemic areas including the neighboring State of Espírito Santo and the Amazon basin with successful results [9,10].

Besides the clinical reports in humans, several eco-epidemiological studies were carried out in Caratinga. At that time, in the 1970s, *Leishmania* (*Viannia*) *braziliensis* was the main ethiological agent of ATL in the region. Interestingly, in some of these strains was detected the presence of LRV1 virus [11], known to increase the severity of the disease [12]. Later, two *Leishmania major* strains were identified among the parasites isolated from patients. This finding was very surprising since this species is commonly found in the Old World and was isolated from rural workers that had never left the region [13]. It is still unknown how *L. major* was introduced in the area and the vector incriminated in its transmission. A sand fly survey

performed in the rural areas detected the presence of *Nyssomyia whitmani* and *Migonemyia migonei* [1]. Although the latter is a permissive vector, its role in the transmission of *L. major* is yet to be determined [14,15]. Regarding reservoirs, the presence of *Leishmania* parasites was found in dogs but not in wild rodents [3]. This is expected since in wild reservoirs the parasitemia is very low and at that time sensitive molecular techniques were not available [16,17]. Altogether, the above-mentioned studies were very important to understand several epidemiological aspects of ATL in the New World. This knowledge was applied to other regions and perhaps countries in Latin America. Unfortunately, in the early 2000s, the Reference Center of Leishmaniasis in Caratinga was closed and ATL studies ceased. The patients continued to be referred to the city public health system. However, an important gap remains in the past two decades since no information on the disease dynamics was reported.

Since the 1990s, geographic information systems (GIS) and special analysis methods have become valuable tools for epidemiological studies in the health field not only in infectious but also in non-infectious diseases [18–21]. Those analyses may help local governments to formulate health policies and monitor affected areas of a given disease [22–27]. In the case of Leishmaniasis, those tools have been used in the last decade for visceral and cutaneous leishmaniasis [28,29]. Due to its occupational and environmental features ATL spatial studies have been reported in Latin America showing specific differences according to the country [30]. In general, ATL transmission is affected by several factors including environmental degradation, work activities, vector proliferation and proximity to water collections [31]. Although most studies have shown a correlation between gender and occupational exposure in several countries [30, 32], this feature may not be clear in others [30]. Although ATL may be related to forested areas, GIS have found potential peridomiciliary and urban areas of transmission in Northern Argentina [32,33]. In Brazil, although most of the ATL cases occurs in the Amazon region, the epidemiological scenarios vary largely in the country. In the Northern state of Pará, the main risk factors associated with ATL transmission were deforestation and sand fly proliferation [27]. In Acre state, the pattern of transmission varies within the state [34]. In some areas occurs intra/peridomiciliary transmission whereas in others it has a forest/sylvatic cycle. In the southeast state of São Paulo, several studies have been carried out in different cities including Teodoro Sampaio, Bauru and Campinas [23,25,29]. Depending on the city, peridomiciliary/urban transmission and/or sylvatic transmission may be observed. This analysis has not been performed in the city of Caratinga, one of the high priority municipalities in Minas Gerais state [35].

Minas Gerais is one of the Brazilian states with higher transmission rates in the country. In 2015, an ecological accident in a mining area resulted in huge contamination of Doce river, the major water supply in the area. This event resulted in serious environmental and economic damage to the local population [36,37]. However, its impact on vector-borne diseases such as leishmaniasis is still uncertain. Recently, two studies have demonstrated that five mesoregions including the North and River Doce Valley, where Caratinga is located are the most seriously affected by ATL [35,38]. These studies showed that several areas in the state of Minas Gerais exhibit a moderate to high risk of transmission in peri-urban areas as a result of ecological features and vector diversity. In this region, the most vulnerable populations included children, pregnant women and indigenous. However, no information on ATL cases and their special distribution have been reported in the past twenty years in the region of Caratinga.

As a wider study on ATL epidemiology in Minas Gerais, this study aimed to analyze the spatial distribution of this disease from 2007 to 2018, sand fly presence and *Leishmania* infection. This information may elucidate the current status of its transmission and help the health authorities to monitor and prevent further ATL outbreaks in the municipality of Caratinga, Minas Gerais, Brazil.

## Materials and methods

### Ethical statement

This Project was approved by the Brazilian Ethical Committee of Research (CEP) under the license #3.997.721.

### Study area

The study was carried out in the municipality of Caratinga (19˚47'25"S and 42˚8'21"W), located in the Rio Doce Valley and belongs to the metropolitan region of Steel Valley, located about 310 km east of the capital Belo Horizonte in the state of Minas Gerais, Brazil. The municipality has a tropical climate and occupies an area of 1,258.479 km$^2$, of which 15.9 km$^2$ are in an urban area, and its population in 2020 was 93,603 inhabitants. In the Gross Domestic Product (US$ 3,410.74), the areas of industry and service provision stand out, however agriculture and livestock also represent a relevant participation, especially with coffee production. Caratinga metro area includes ten districts: Cordeiro de Minas, Dom Lara, Dom Modesto, Patrocínio de Caratinga, Santa Efigênia de Caratinga, Santa Luzia de Caratinga, Santo Antônio do Manhuaçu, Sapucaia, São Cândido and São João Jacutinga (Fig 1).

### Data collection and spatial analysis

First, the number of ATL cases (2007–2018) was evaluated based on the data from the municipal health service. Identification of patients was kept confidential. Demographic information including gender, age, educational level, clinical form, diagnostic criteria and case evolution were assessed. Then, to analyze their spatial distribution they were grouped into 4 three-year intervals (2007–2009; 2010–2012; 2013–2015 and 2016–2018). These intervals minimize the ATL fluctuations occurring each year as reported elsewhere [35].

**Bayesian analysis.** The incidence rates were re-estimated for each of the geographic analytical units and for each three-year period, using Bayesian empirical spatial smoothing. After calculating the incidence rate, the accumulated smoothed incidence rate (three-year period) (Bayesian spatial smoothing) was calculated for each demographic sector. A first-order adjacency matrix was created, and the smoothing was performed to reduce random fluctuation. This facilitates the subsequent analysis of spatial data, since areas with small populations and few cases may imply great variation in their rates [35]. This analysis used GeoDa software version 1.14 (ASU, GeoDa Center for Geospatial Analysis and Computation, Arizona, USA).

**Directional Distribution Ellipses** provides a spatial distribution of events in two directions: cluster identification and orientation. The largest axis defines the direction of maximum distribution dispersion, while the smallest is perpendicular to the previous axis and defines the minimum dispersion [39]. The application of directional distribution was employed using the Q GIS Software version 3.4.14 'Madeira'.

**Kernel Density Maps** were used to identify the level of clusters, hotspots, of the ATL cases. Each observation is weighted according to a central distance. This creates a continuous surface that represents where the densities are located [40]. QGIS software version 3.4.14 'Madeira' was used for determining Kernel density maps using an influence radius of 600m.

**Scanning analysis of space-time clusters.** This analysis was performed by gradually scanning information in space and time, indicating the number of events observed and expected within each unit of analysis [41,42]. To identify spatio-temporal clusters, scanning statistics were applied, using the SaTScan 9.4.4 software [42].

The identification of spatio-temporal clusters was made using the Poisson discrete model [42,43]. The spatio-temporal scanning techniques were configured to detect clusters of high

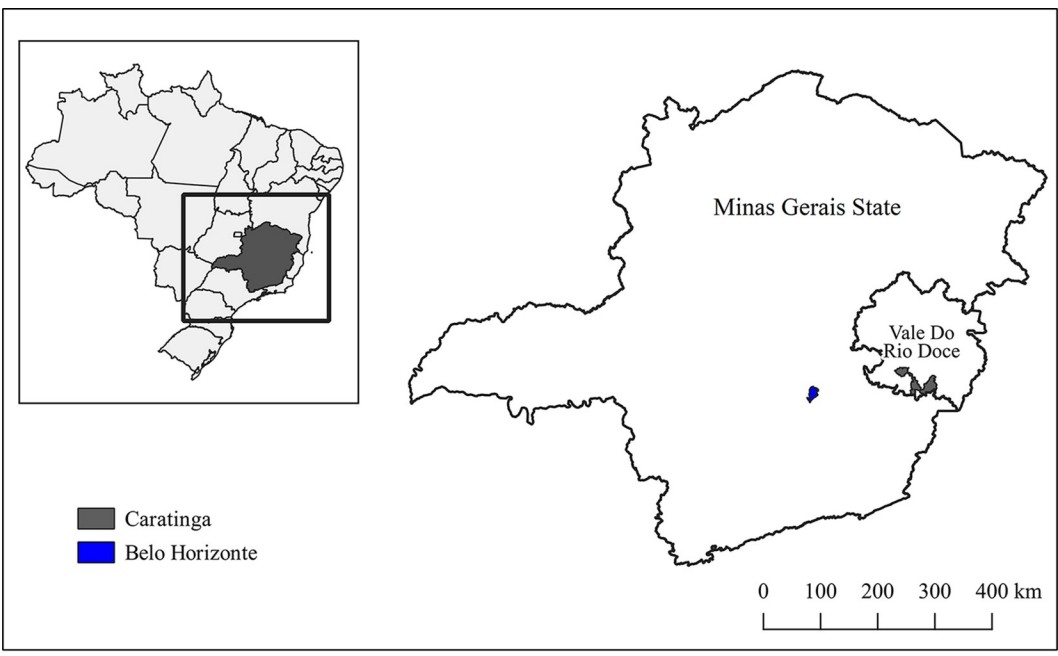

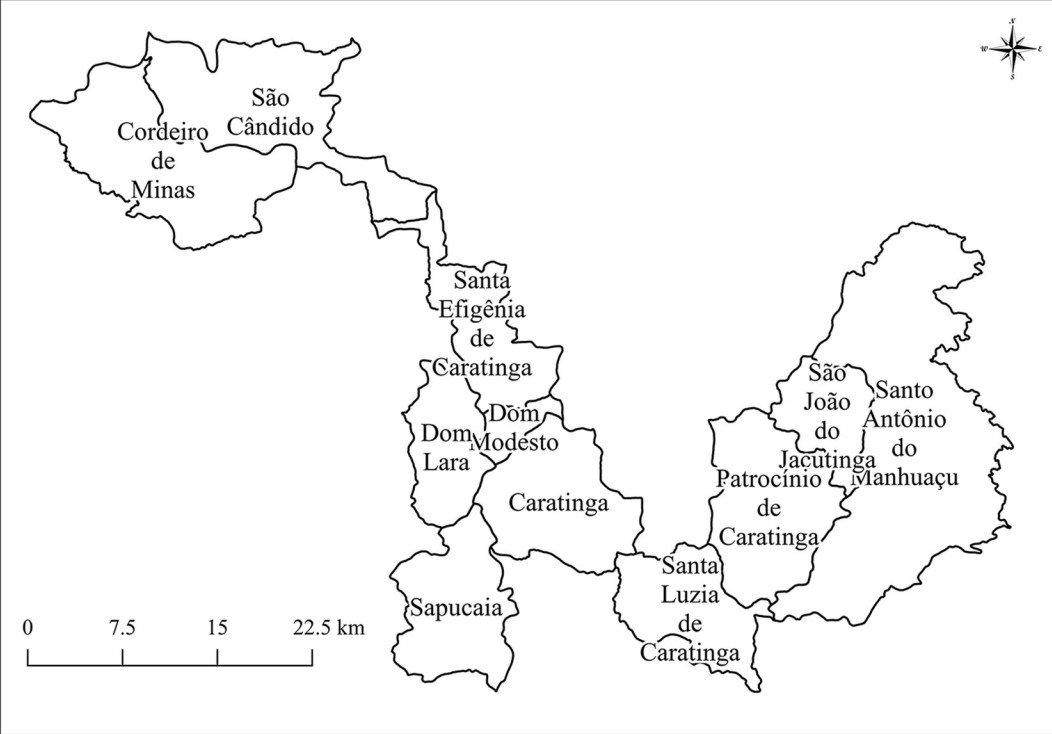

**Fig 1. Location of the municipality of Caratinga and their districts in the state of Minas Gerais, Brazil (www.ibge.gov.br).**

and low risk for the transmission of ATL. The significance test of the identified clusters was based on the comparison of a null distribution obtained by Monte Carlo simulation [41,42] and relative risks (RR) of each cluster were determined.

**Spatial regression.** Ordinary Least Square (OLS) model, a global spatial analytical method was performed [44]. Population, garbage dumped in wasteland, income up to ½ minimum

wage, afforestation in the streets and residence without sewage collection were included as explanatory variables. This analysis generates the Akaike Information Criterion (AIC). Those data for each census sector was obtained from the Brazilian Institute of Geography and Statistics (IBGE) until 2010 [45].

**Sand fly captures.** Sand fly captures were authorized by Ministry of Environment of Brazil (protocol # 15237–2). Based on the previous results, sand flies were captured in the most affected areas for ATL. Two captures (July and September 2020) were performed in four neighborhoods of Caratinga (Anápolis, Esplanada, Downtown and Limoeiro) and in the rural areas (Patrocínio de Caratinga and Sapucaia). The captures were performed using HP traps [46] for two nights. Captured insects were conditioned in a 50mL Falcon Tube containing glycerin alcohol prior to taxonomic identification.

*Leishmania* **detection.** After sex screening, females were subjected to DNA extraction for *Leishmania* detection using PCR amplification of Internal Transcribed Spacer 1 (ITS1) [47]. In all reactions, a negative control (sterile water as template), positive control with 20 nanograms of DNA extracted from reference strains of *L. amazonensis* (IFLA/BR/67/PH8), *L. braziliensis* (MHOM/BR/75/M2903), *L. infantum* (MHOM/BR/74/PP75), *L. guyanensis* (MHOM/BR/75/M4147) and *L. major* (MHOM/IL/81/Friedlin), were used. Positive samples were submitted to digestion by the restriction enzyme *Hae*III to identify *Leishmania* species [48].

## Results

Notification records (319) were retrieved from the Caratinga Health Service from January 2007 to December 2018. Males (57.37%) represented the majority of the ATL cases followed by females (42.63%) (183 versus 136). Age averages were 33.26 ±19.87 and 35.19 ± 19.87 for men and women, respectively. Most of the individuals analyzed (79.94%) had a low level of education. Localized cutaneous lesions were the most prevalent form of the disease (97.49%). ATL cases (84.01%) were concentrated in the countryside and 94.04% have evolved to cure after Glucantime therapy (Table 1).

ATL cases peaked between 2009–2011 and increased after 2015 (Fig 2). All cases (100%) were notified as being from Caratinga. However, in some records (101) this information was not assessed. During the years 2010 and 2018, the highest ATL numbers were recorded. Consistent with this information the total incidence rates per 10,000 inhabitants also followed this pattern, ranging from 0.72 in 2007 to 6.57 in 2010, year with the highest incidence. The average incidence in the municipality of Caratinga was 3.01 (Table 2). The district of Patrocínio de Caratinga contributed with 45.77% of the cases. The other districts contributed with a smaller number of cases, with eight cases in Dom Modesto (2.51%), four cases in Cordeiro de Minas and Dom Lara (1.25%) and two cases in São Cândido (0.63%) (Table 3).

In general, after Bayesian analysis (Fig 3)., the smoothed incidence was lower in the 2007–2009 period compared to the others. During this time, no incidence above 100/10,000 inhabitants was detected and ATL cases were absent in some districts including Cordeiro de Minas and São Cândido. In the subsequent periods, this incidence increased above 100/10,000 inhabitants including the urban area of Caratinga (2010–2012) and the rural areas of Patrocínio de Caratinga (2010–2018), Santo Antônio do Manhuaçu (2010–2012; 2016–2018) and Sapucaia (2016–2018). When the same analysis was applied to the urban area of Caratinga we also detected an increase in the incidence in the northeast part of the city (2010–2012) and in the southwest (2013–2018) (Fig 4).

In general, directional distribution ellipses showed that the spatial distribution of ATL cases in Caratinga remained unchanged between 2007–2018. In the first three years (2007–2009), the ellipse comprised cases located in the rural areas. After 2010, the ellipses shift towards urban areas of the municipality (Fig 5).

**Table 1. Demographic and clinical features of ATL cases in the municipality of Caratinga (2007–2018).**

| Feature | | n (%) | % |
|---|---|---|---|
| **Gender** | | | |
| Male | | 183 | 57.37 |
| Female | | 136 | 42.63 |
| **Age group (years) by gender** | Male | Female | |
| <1 | 1 (0.55) | 0 (0) | 0.31 |
| 1–10 | 18 (9.84) | 19 (13.97) | 11.60 |
| 11–20 | 46 (25.14) | 19 (13.97) | 20.38 |
| 21–30 | 27 (14.75) | 18 (13.24) | 14.11 |
| 31–40 | 25 (13.66) | 28 (20.59) | 16.61 |
| 41–50 | 22 (12.02) | 23 (16.91) | 14.11 |
| 51–60 | 23 (12.57) | 11 (8.09) | 10.66 |
| 61–70 | 14 (7.65) | 12 (8.82) | 8.15 |
| 71–80 | 7 (3.83) | 3 (2.21) | 3.13 |
| ≥81 | 0 (0) | 3 (2.21) | 0.94 |
| **Education level** | | | |
| Illiterate | | 8 | 2.51 |
| Incomplete elementary school | | 243 | 76.18 |
| Complete primary education | | 4 | 1.25 |
| High school | | 2 | 0.63 |
| No school age | | 19 | 5.96 |
| Uninformed | | 31 | 9.72 |
| Ignored | | 12 | 3.76 |
| **Clinical form** | | | |
| Cutaneous | | 311 | 97.49 |
| Mucocutaneous | | 8 | 2.51 |
| **Case outcome** | | | |
| Clinical cure | | 300 | 94.04 |
| Abandonment treatment | | 7 | 2.19 |
| Death of other cause | | 1 | 0.31 |
| Transferred | | 1 | 0.31 |
| Not informed | | 10 | 3.13 |
| **Area of notification** | | | |
| Urban | | 42 | 13.17 |
| Periurban | | 9 | 2.82 |
| Rural | | 268 | 84.01 |

Kernel density maps detected several hotspots including the district of Patrocínio de Caratinga (all years), the urban area of Caratinga city (all years) and the district of Sapucaia (2007–2009, 2010–2012 and 2016–2018) (Fig 6A–6D). To confirm if the human cases were occurring in the urban area, we repeated the analysis excluding all records whose exact addresses were missing. This selection enabled the analysis of 39 cases (Fig 7). Although some cases were observed between 2007 and 2009, after 2010, an increase in the hotspots was detected in the central and southwest parts of the city.

To refine our previous findings and determine the spatial relative risk (RR), we performed cluster scan analysis including other variables. This analysis identified two areas (low and high) risk for ATL transmission (Fig 8). The low-risk cluster was the urban part of Caratinga,

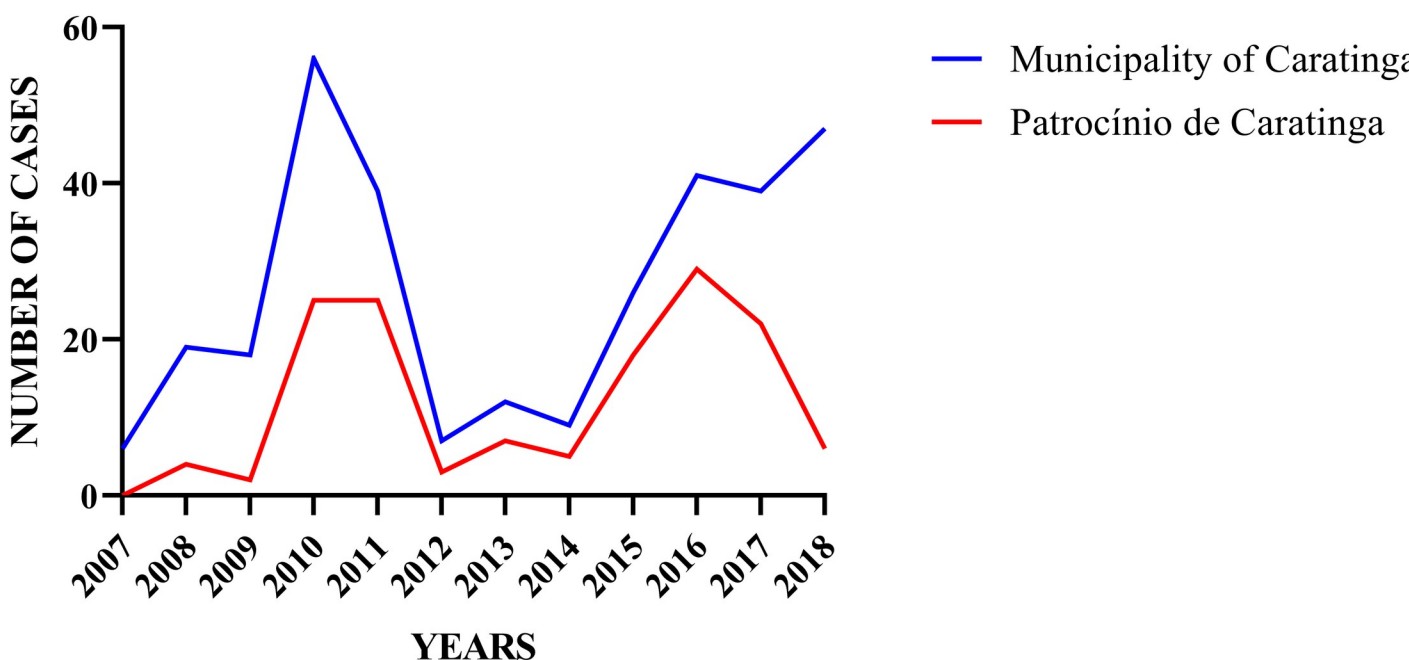

**Fig 2. Distribution of ATL cases in the city of Caratinga and in the district of Patrocínio de Caratinga.**

with a relative risk (RR) of 0.1 in the period from 2011 to 2016. The high-risk cluster encompassed the rural area, this cluster had a RR of 27.6 in the period from 2013 to 2018 (Fig 8).

To detect if demographic aspects of the region could affect the spatial distribution of the cases, a regression analysis (OLS) was performed in the entire study area. The variables income of up to half a minimum wage and garbage dumped in vacant lots affected ATL incidence (P<0.05) (Table 4). In the urban area, trees in the street were the only variable affecting ATL incidence (P <0.05) (Table 5).

To investigate the presence of sand flies and perhaps, *Leishmania* infection, a total of 113 insects were captured from July to September. Fifty-nine females (52.21%) and 54 males

**Table 2. Number of cases, population and incidence of ATL cases in the municipality of Caratinga (2007–2018).**

| Year | n | Population | Incidence* |
|---|---|---|---|
| 2007 | 6 | 83,363 | 0.72 |
| 2008 | 19 | 84,825 | 2.24 |
| 2009 | 18 | 85,469 | 2.11 |
| 2010 | 56 | 85,239 | 6.57 |
| 2011 | 39 | 85,811 | 4.54 |
| 2012 | 7 | 86,364 | 0.81 |
| 2013 | 12 | 89,578 | 1.34 |
| 2014 | 9 | 90,192 | 1.00 |
| 2015 | 26 | 90,782 | 2.86 |
| 2016 | 41 | 91,342 | 4.49 |
| 2017 | 39 | 91,841 | 4.25 |
| 2018 | 47 | 91,503 | 5.14 |
| Total | 319 | Average | 3.01 |

*Per 10,000 inhabitants

**Table 3. Number of ATL cases by city/district during the period from 2007 to 2018.**

| City/district | n | % |
| --- | --- | --- |
| Caratinga | 63 | 19.75 |
| Cordeiro de minas | 4 | 1.25 |
| Dom Lara | 4 | 1.25 |
| Dom Modesto | 8 | 2.51 |
| Patrocínio de Caratinga | 146 | 45.77 |
| Santa Efigênia | 18 | 5.64 |
| Santa Luzia | 11 | 3.45 |
| Santo Antônio do Manhuaçu | 19 | 5.96 |
| São Cândido | 2 | 0.63 |
| São João do Jacutinga | 15 | 4.70 |
| Sapucaia | 29 | 9.09 |

(47.79%) were identified belonging to four species: *Ny. intermedia* (13.27%), *Ny. whitmani* (82.30%), *Evandromyia* complex *cortelezzii* (1.77%) and *Mi. migonei* (2.65%) (Table 6). None of the 59 captured females tested positive for the presence of *Leishmania* DNA.

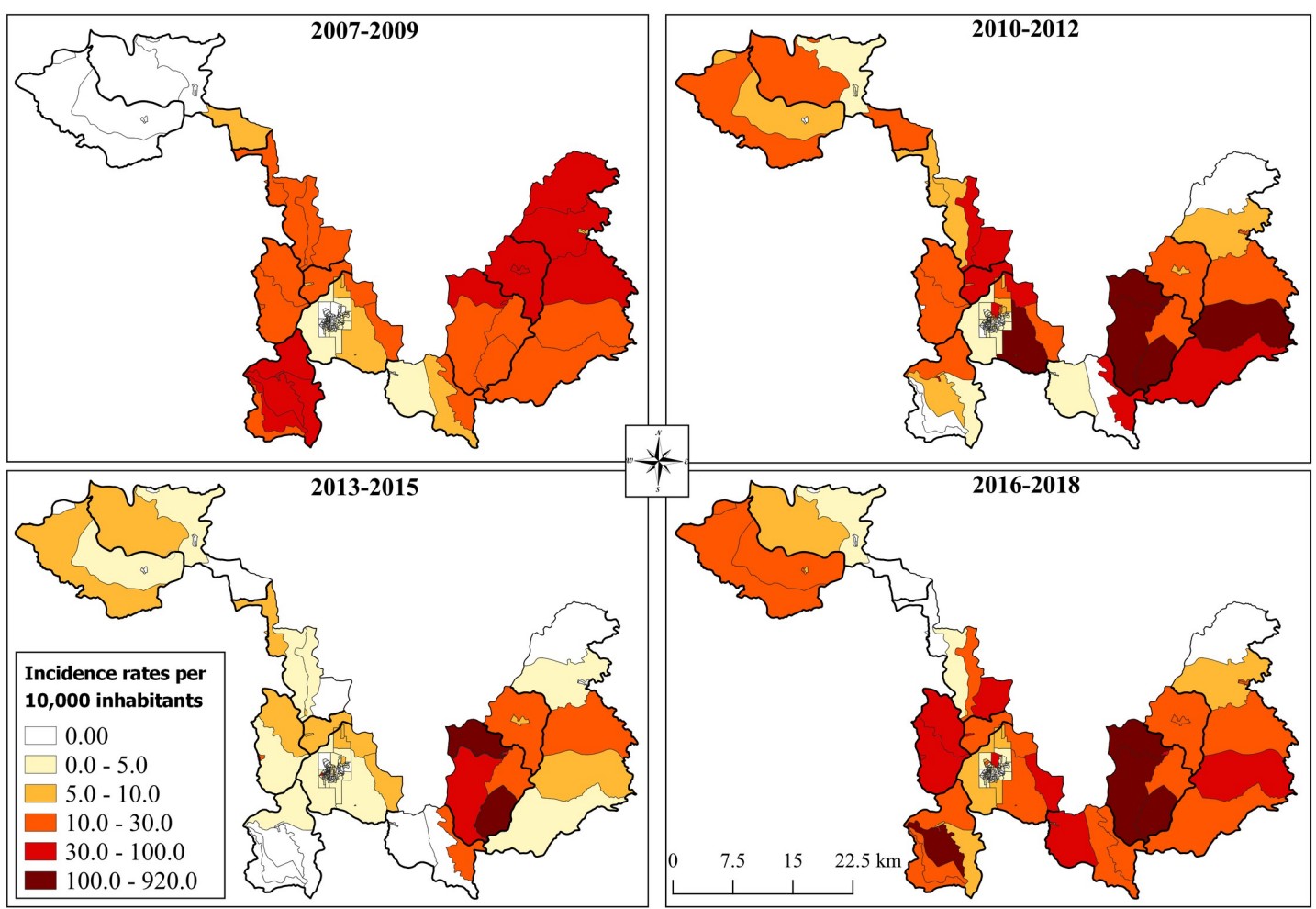

**Fig 3. Smoothed Accumulated Incidence, per 10,000 inhabitants, of ATL cases belonging to the municipality of Caratinga (www.ibge.gov.br).**

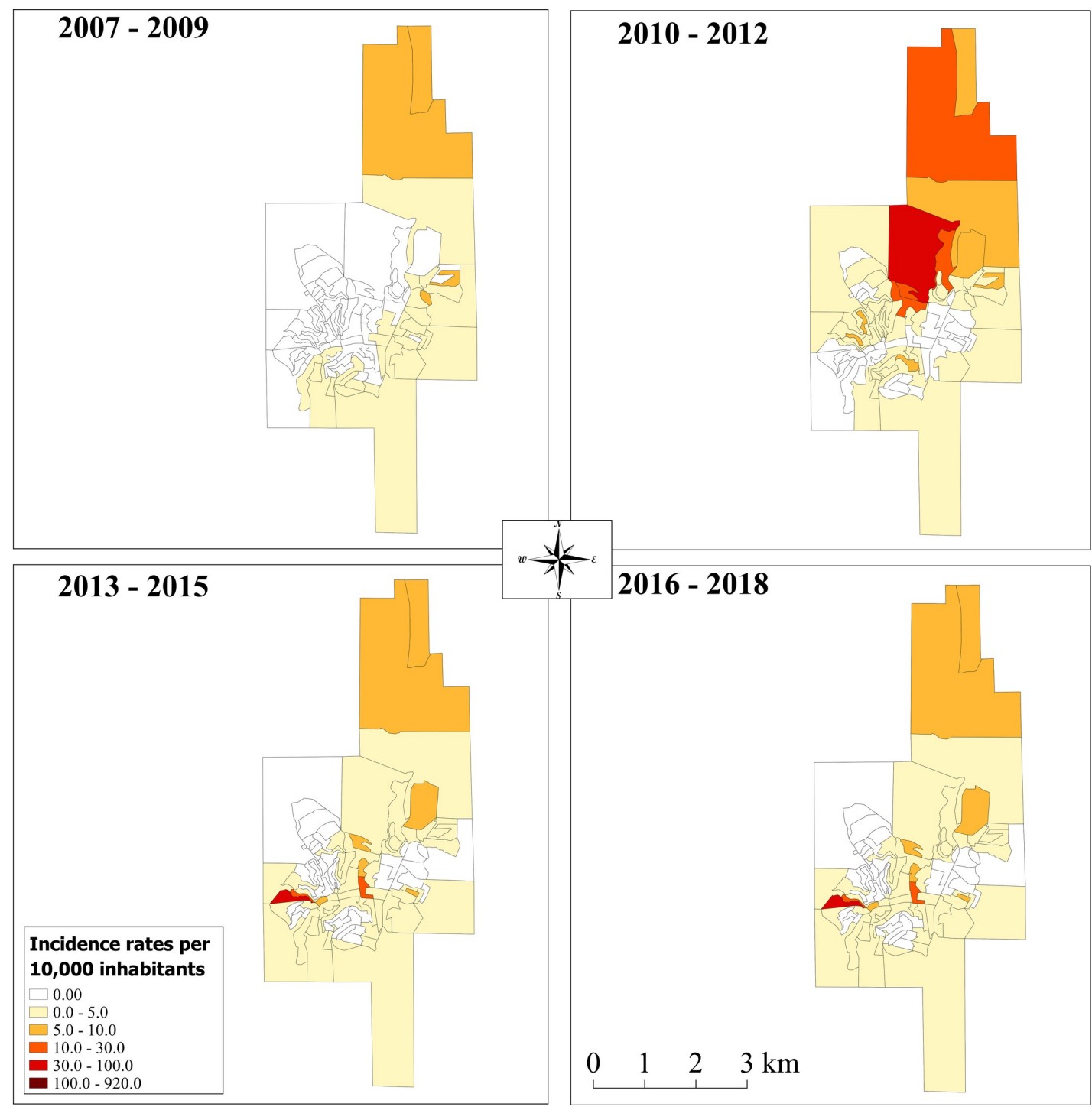

**Fig 4. Smoothed Accumulated Incidence, per 10,000 inhabitants, of ATL cases belonging to the urban area of Caratinga ([www.ibge.gov.br](www.ibge.gov.br)).**

## Discussion

The municipality of Caratinga is included in one of the mesoregions most affected by ATL in the state of Minas Gerais [35]. Previous reports in the Brazilian states of Paraná [24], São Paulo [29], Amazonas [49] and Minas Gerais [38] showed that demographically ATL is primarily a

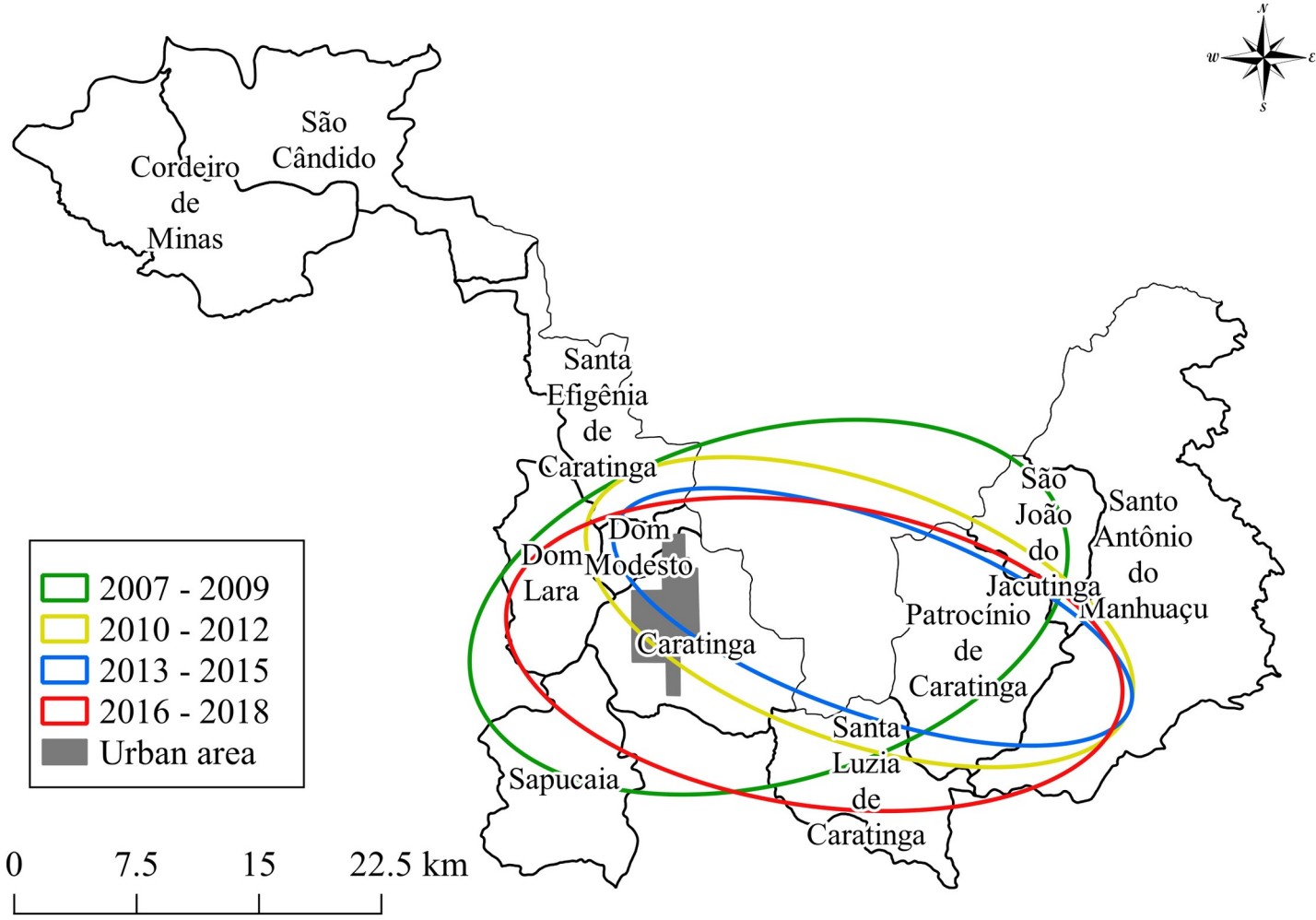

**Fig 5. Ellipse showing the distribution of cases of ATL over a period of three years during the period 2007–2018 ([www.ibge.gov.br](http://www.ibge.gov.br)).**

rural disease affecting males and low educated individuals, a profile confirmed by our demographic analysis. Our results, together with those of Melo et al., 2017 [24], showed that the cutaneous leishmaniases is the most prevalent form (~98%) and chemotherapy is the main prophylactic control measure. It is interesting to notice that despite the fact that most cases are from rural origin, a considerable number of cases were notified as urban based on Bayesian and Kernel analysis. Most of the rural cases were found mainly in the districts of Patrocínio de Caratinga and Sapucaia. However, a high frequency of urban cases was detected, and two reasons may be considered in this scenario: 1) those patients live in the urban area and frequently travel to the surrounding rural areas, a common situation since it is very usual to have a cabin in the countryside and, 2) transmission could also be occurring in the urban area. There was no homogeneity in the incidence rates in the evaluated period. The low incidence rates observed between those peaks could be a result of several notification-derived factors including underreporting. This is a constant issue not only in Brazil, but also in several other endemic places including Argentina and Colombia [50–55]. After 2015, an increase in the ATL incidence was noticed, probably due to diagnosis improvement, disease transmission increase or both. This also confirms previous findings in other Brazilian endemic regions [56–58] and could be justified by anthropic actions, socioeconomic pressure, and disordered urbanization.

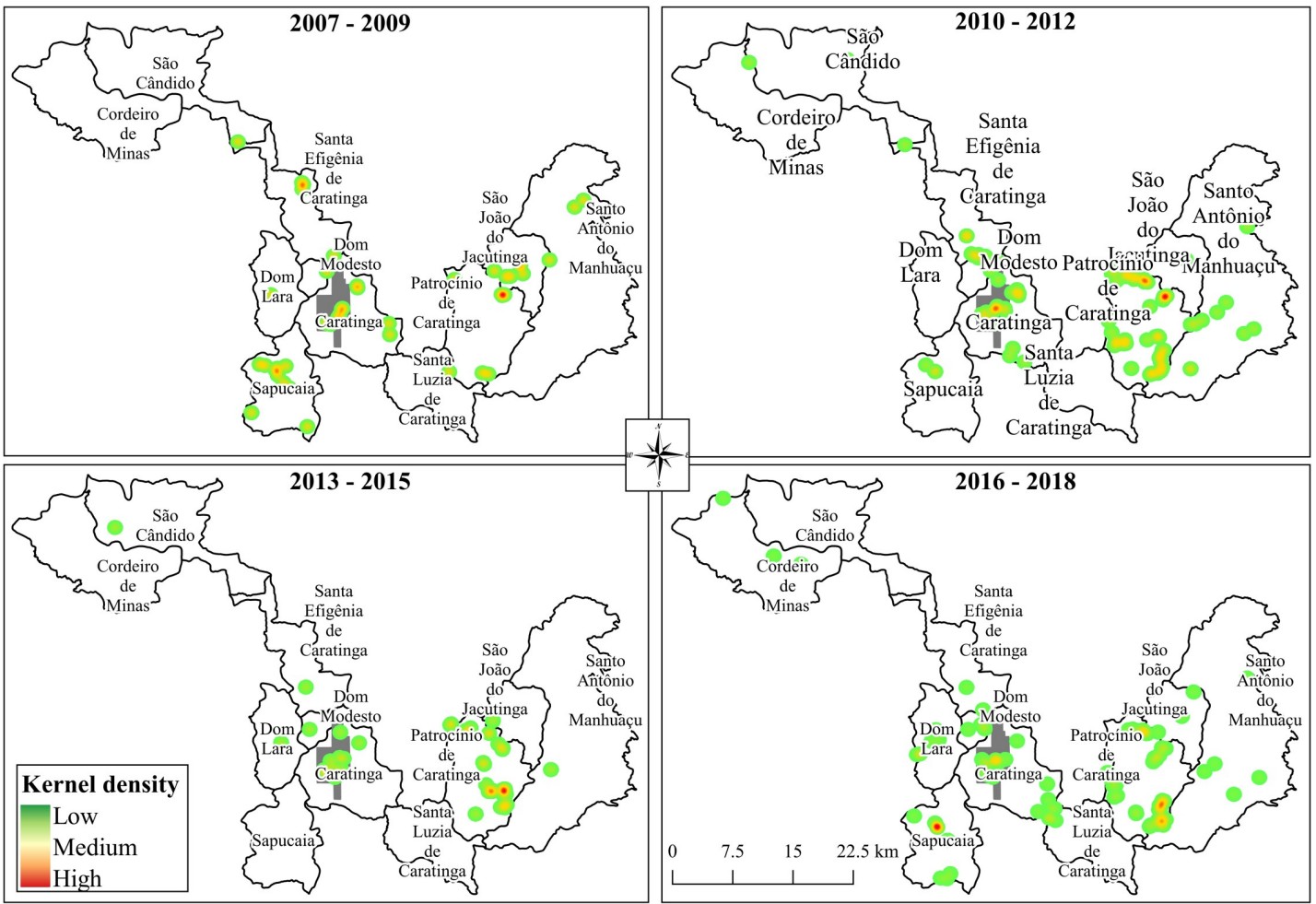

**Fig 6. Kernel density of ATL cases in the municipality of Caratinga ([www.ibge.gov.br](www.ibge.gov.br)).**

These phenomena may contribute to sand fly distribution and, consequently for ATL transmission. However, it is very clear a shift in the distribution of cases caused by an increase in the number of ATL cases in Sapucaia especially between 2016–2018. Those data reinforce that together with Patrocínio de Caratinga, Sapucaia must undergo stricter surveillance since in rural districts, this increase may be related to agricultural expansion and proximity to sand flies [59–61].

It is important to notice that the urban area of Caratinga has always shown hotspots in the entire period. In Brazil, urbanization is often reported for visceral leishmaniasis, where major cities have become endemic areas for *Leishmania infantum* [28,62,63]. As we mentioned before, the constant presence of urban cases in Caratinga deserves more attention and may indicate recent urbanization of ATL. The residents may have acquired the disease not only in the countryside but also in the city. The hotspot located in the north part of the city is very urbanized and close to a quarry. The hotspot located in the southwest is primarily a residential area. After relative risk (RR) analysis, the city of Caratinga and district of Patrocínio de Caratinga represents low and high areas for ATL transmission, respectively. This reinforces the need to prioritize ATL surveillance and control measures in these locations, especially where the highest risk is concentrated [64–67].

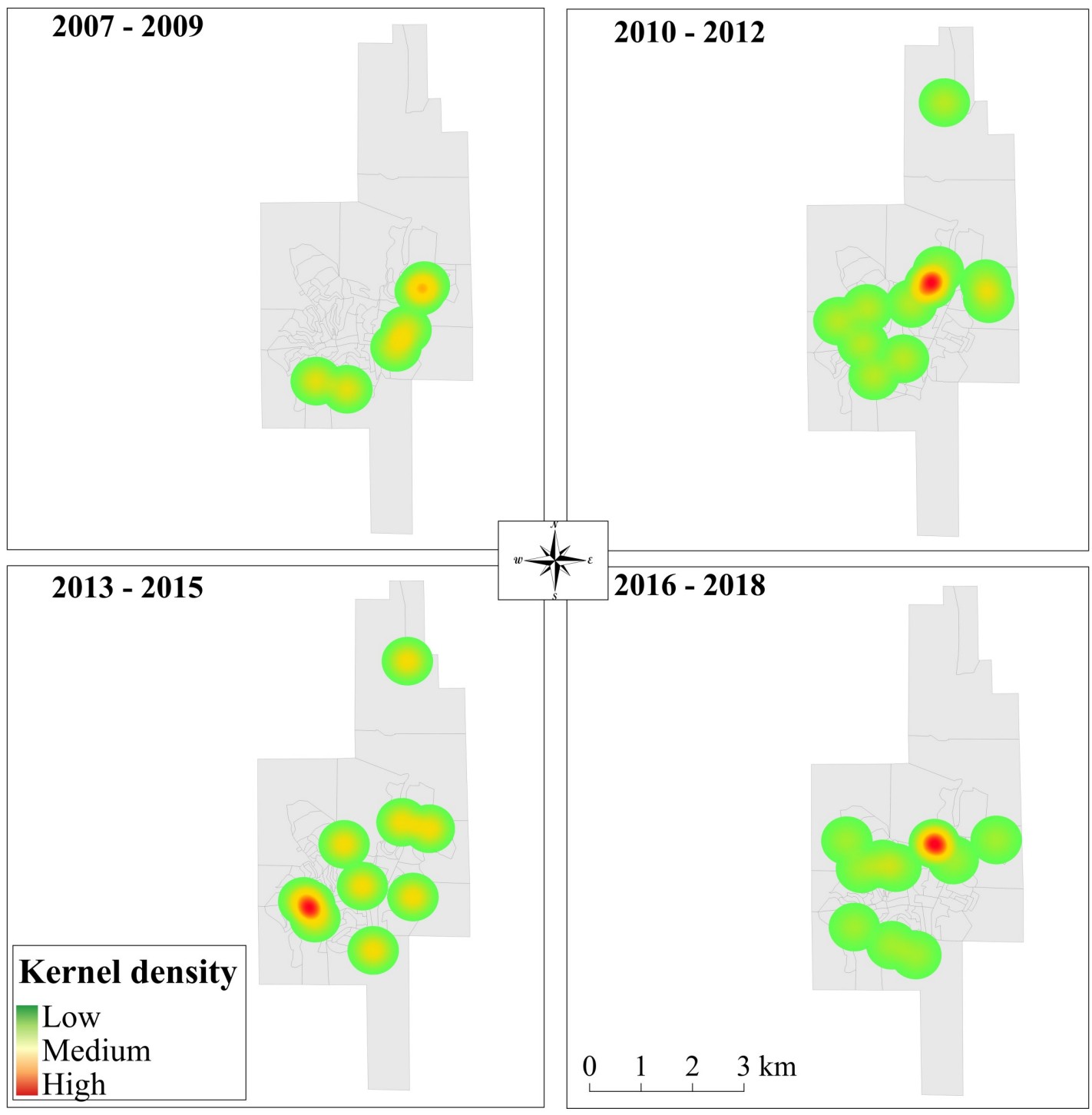

**Fig 7. Kernel density of cases of ATL in the city of Caratinga ([www.ibge.gov.br](http://www.ibge.gov.br)).**

After OLS analysis, 56% of ATL cases in Caratinga can be explained by income and waste disposal in vacant lots. Our results suggest that ATL is related to poverty and lack of basic sanitation conditions in the entire municipality of Caratinga. In the urban area, 5% of ATL cases can be explained by the presence of trees in the streets of the residences. Although only 5% of

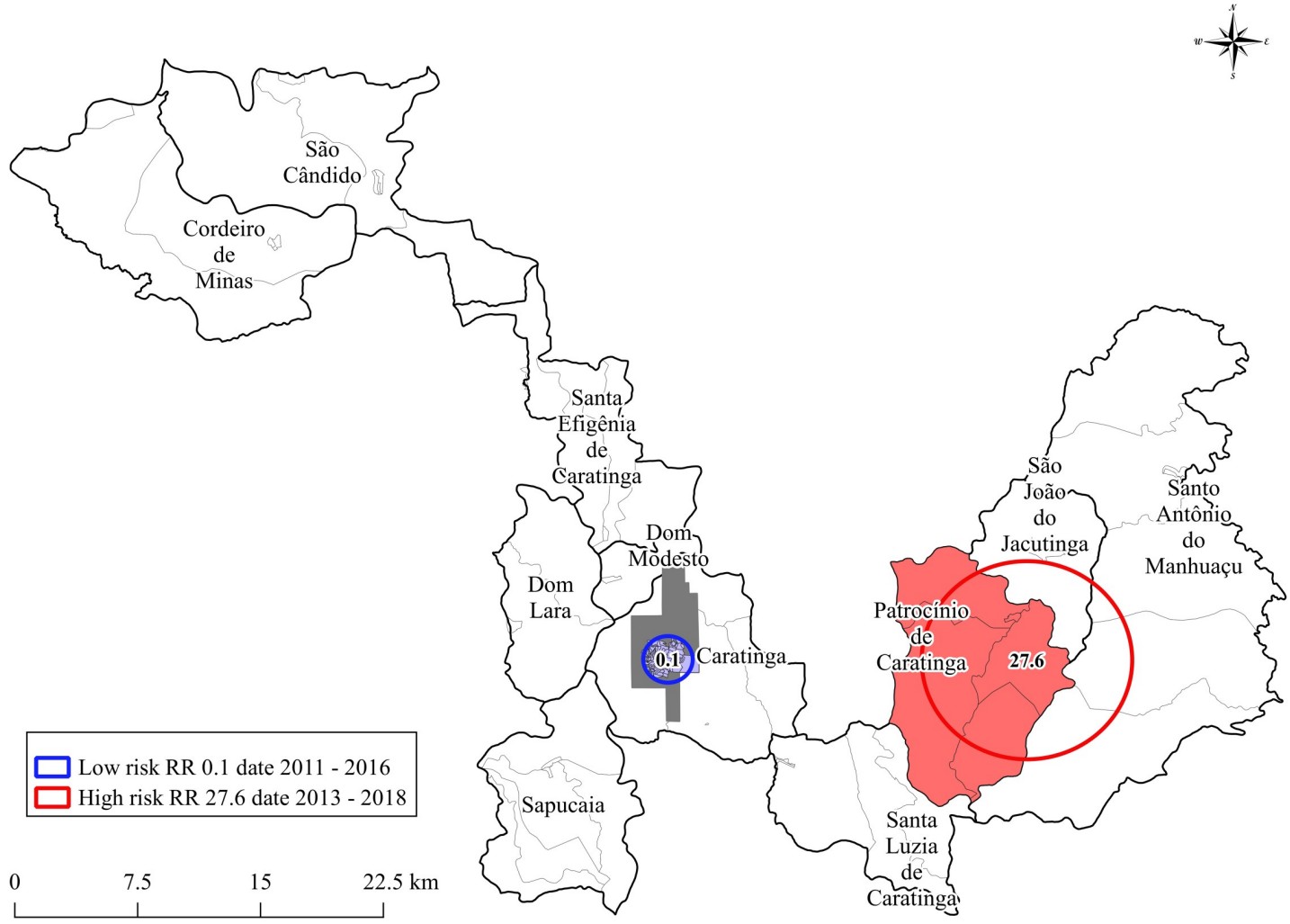

**Fig 8. Relative risk of ATL cases reported in the city of Caratinga and in the district of Patrocínio de Caratinga (www.ibge.gov.br).**

cases in the urban area can be explained by our spatial regression model, the presence of trees on the sidewalk showed statistical significance, which may indicate in neighborhoods with trees may have a higher incidence of ATL in the urban area. This same variable was not

**Table 4. Ordinary Least Squares (OLS) Regression of leishmaniasis incidence rates by census sector in Caratinga area (all districts).**

| Variables | Coefficient | t-Statistic | P | Log likelihood | AIC* |
|---|---|---|---|---|---|
| Constant | -42.7349 | -2.6108 | 0.010 | -837,871 | 1,685.74 |
| Waste disposal on vacant lots | 8.5288 | 8.8905 | 0.001 | | |
| Income up to ½ minimum wage | 1.2428 | 4.4292 | 0.001 | | |
| Trees in the street | 0.2255 | 1.4531 | 0.148 | | |
| Residence without sewage collection | 0.1717 | 1.0597 | 0.291 | | |
| | R2 adjusted | | p-Value F-statistic | | |
| | 0.561 | | p = 0.01 | | |

*Akaike information criterion.

**Table 5. Ordinary Least Squares (OLS) Regression of leishmaniasis incidence rates in the urban area of Caratinga.**

| Variables | Coefficient | t-Statistic | P | Log likelihood | AIC* |
|---|---|---|---|---|---|
| Constant | 2.3781 | 0,4093 | 0.68321 | -420,365 | 850,739 |
| Waste disposal on vacant lots | -0.1780 | -0,1026 | 0.9184 | | |
| Income up to ½ minimum wage | 0.0566 | 2,0549 | 0.62 | | |
| Trees in street | 0.1095 | 2,0549 | 0.042 | | |
| Residence without sewage collection | 0.0931 | 1,4009 | 0.164 | | |
| | R2 adjusted | | p-Value F-statistic | | |
| | 0.052098 | | p = 0.01 | | |

*Akaike information criterion.

statistically significant when comparing the entire municipality. The comprehension of such factors may be useful to explain the transmission patterns of ATL in countryside municipalities and their understanding could be helpful for the improvement of public policies regarding disease control. However, it is worth mentioning that the data used to make such analysed were obtained from the demographic census of the IBGE [45], carried out in 2010, which implies the difficulty of relating other variables to the real situation of ATL in the municipality.

This study identified the phlebotomine sand fly fauna in the urban areas of the city of Caratinga and the districts of Patrocínio de Caratinga and Sapucaia. In the municipality of Caratinga, and in the other areas assessed, ATL cases and environmental impacts have historically been related to human occupation processes. The presence of the phlebotomine sand flies species found confirms the ability of these insects to adapt to environments different from their natural habitats [68,69]. It is important to relate the vector density to environmental aspects favorable to peridomestic sand flies, such as the presence of vegetation, roots, tree trunks and organic material, which are the possible shelters and breeding sites [70].

In the locations where the samples were collected, except for the center of Caratinga, there are mainly simple households, with poor basic sanitation and a large population of domestic animals. These conditions were more evident in Patrocínio de Caratinga and Sapucaia, where a high frequency of sand flies captured was found compared to some neighborhoods in Caratinga. In these neighborhoods, families kept different types of livestock outside the home as a means of subsistence. It is known that these factors associated with the low economic condition of residents contribute considerably to the transmission of ATL [71]. Here, a total of 113

**Table 6. Sandflies collected by sex and location in the urban area of Caratinga, Patrocínio de Caratinga and Sapucaia in July and September 2020.**

| Species | Caratinga | | | | | | | | Patrocínio de Caratinga | | Sapucaia | | Total (%) | | |
|---|---|---|---|---|---|---|---|---|---|---|---|---|---|---|---|
| | Anápolis | | Esplanada | | Downtown | | Limoeiro | | Center | | Center | | | | |
| | ♀ | ♂ | ♀ | ♂ | ♀ | ♂ | ♀ | ♂ | ♀ | ♂ | ♀ | ♂ | ♀ | ♂ | |
| *Evandromyia* complexo *cortelezzii* | 0 | 0 | 0 | 0 | 0 | 0 | 0 | 0 | 0 | 0 | 2 | 0 | 2 (100) | 0 (0.00) | 2 (1.77) |
| *Migonemyia migonei* | 0 | 0 | 0 | 0 | 0 | 0 | 0 | 0 | 1 | 0 | 1 | 1 | 2 (66.67) | 1 (33.33) | 3 (2.65) |
| *Nyssomyia intermedia* | 1 | 1 | 1 | 2 | 0 | 0 | 0 | 0 | 3 | 4 | 1 | 2 | 6 (40.00) | 9 (60.00) | 15 (13.27) |
| *Nyssomyia whitmani* | 8 | 9 | 14 | 10 | 0 | 0 | 7 | 7 | 8 | 9 | 12 | 9 | 49 (52.69) | 44 (47.31) | 93 (82.30) |
| Total (%) | 9 | 10 | 15 | 12 | 0 | 0 | 7 | 7 | 12 | 13 | 16 | 12 | 59 (52.21) | 54 (47.79) | 113 (100) |
| | 19 (16.81) | | 27 (23.89) | | 0 (0.00) | | 14 (12.39) | | 25 (22.12) | | 28 (24.78) | | 113 (100) | | |

sand flies were captured and the most abundant species found was *Ny. whitmani* (82.30%), a proven vector of *L. braziliensis*. In some regions of Brazil this species is not yet urbanized or is in the process of urbanization, while in other regions the species is well adapted to the urban environment, as reported elsewhere [72]. The identification of the natural infection rates of sand flies by *Leishmania* is essential to determine the risk of infection for the hosts of an area, allowing adequate planning in the prevention and control of ATL. The finding of this species in all three areas with ATL patients led us to perform molecular prospection of *Leishmania* parasites in those sand fly vectors. Several epidemiological studies have assessed *Leishmania* infection in sand flies [73–77]. In those studies, the frequency of infection ranged from 0.4% [76] to 39.6% [78]. Here, different from those studies, we did not detect infection by *Leishmania* species, and this may be a result of the low number of collected insects since natural infection is expected to be very scarce [76]. However, since our captures occurred in urban areas closely related to human ATL cases, the presence of proven vectors is a strong indication of disease transmission in Caratinga, Patrocínio de Caratinga and Sapucaia districts [79–81].

The analyses performed in this study to investigate the spatial and temporal distribution of ATL in Caratinga must be interpreted considering the possible limitations imposed using data collected under passive surveillance. Spatial analysis was carried out based on secondary data, leading to studies of epidemiological factors that may present bias. It is known that an important bias that occurs in this type of study is underreporting, potentially due to diagnostic errors, inadequate medical records, failure of the sick person to seek medical attention or even a deficiency in the local surveillance system. The use of other variables such as demographic data associated with secondary data can minimize or eliminate bias in underreporting.

We do not know in which extent the discontinuity of the reference center services could have impacted the increase in the number of cases in the past two decades. However, we could not discharge the possibility that a myriad of factors may be contributing for this including environmental changes, access to health services and lack of knowledge on leishmaniasis. However, this study demonstrated that it is possible to provide, in retrospect, an overview of epidemiological patterns in the municipality using notification data, emphasizing the importance of passive research as a tool for the management of the ATL control program.

This is a multidisciplinary study using either GIS and sand fly assessment in the municipality of Caratinga. In conclusion, ATL cases are increasingly reported during the surveyed period not only in the rural (Sapucaia and Patrocínio de Caratinga districts) but also in two urbanized areas of Caratinga. Although we could not detect *Leishmania* in the sand flies, the presence of four proven ATL vectors overlapping with human cases strongly suggests that those areas are at risk of transmission and should undergo surveillance by the health authorities. Altogether those data confirmed the urbanization of ATL in Caratinga.

## Acknowledgments

To Jacqueline Marli dos Santos and Cicera Dulce Salgado from the Caratinga Public Health Services and the director of the Department of Epidemiological Surveillance José Calos Damasceno for providing ATL records helping us to get geographic data from notifications and the health agent Everaldo Alberto Oliveira for help during sand fly collections. We also thank the Print-Capes Program.

## Author Contributions

**Conceptualization:** Célia M. F. Gontijo, David S. Barbosa, Rodrigo P. Soares.

**Formal analysis:** Rafael L. Neves, Diogo T. Cardoso, Felipe D. Rêgo, Célia M. F. Gontijo, David S. Barbosa, Rodrigo P. Soares.

**Investigation:** Rafael L. Neves, Diogo T. Cardoso, Felipe D. Rêgo, Célia M. F. Gontijo, David S. Barbosa, Rodrigo P. Soares.

**Methodology:** Diogo T. Cardoso, Felipe D. Rêgo, David S. Barbosa.

**Supervision:** Célia M. F. Gontijo, Rodrigo P. Soares.

**Writing – original draft:** Rafael L. Neves, Rodrigo P. Soares.

**Writing – review & editing:** Diogo T. Cardoso, Felipe D. Rêgo, Célia M. F. Gontijo, David S. Barbosa.

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
