## [Decision Letter · Decision Letter 0]

16 Feb 2021

Dear Dr Soares,

Thank you very much for submitting your manuscript "A follow-up study (2007-2018) on American Tegumentary Leishmaniasis cases in the municipality of Caratinga, Minas Gerais State, Brazil an area of recent urban transmission" for consideration at PLOS Neglected Tropical Diseases. As with all papers reviewed by the journal, your manuscript was reviewed by members of the editorial board and by several independent reviewers. In light of the reviews (below this email), we would like to invite the resubmission of a significantly-revised version that takes into account the reviewers' comments. 

We cannot make any decision about publication until we have seen the revised manuscript and your response to the reviewers' comments. Your revised manuscript is also likely to be sent to reviewers for further evaluation.

Sincerely,

Shan Lv, Ph.D.

Deputy Editor

Shan Lv

Deputy Editor

Reviewer's Responses to Questions

**Key Review Criteria Required for Acceptance?**

**Methods**

-Are the objectives of the study clearly articulated with a clear testable hypothesis stated?

-Is the study design appropriate to address the stated objectives?

-Is the population clearly described and appropriate for the hypothesis being tested?

-Is the sample size sufficient to ensure adequate power to address the hypothesis being tested?

-Were correct statistical analysis used to support conclusions?

-Are there concerns about ethical or regulatory requirements being met?

Reviewer #1: p6, line 188: If the relative risks were already estimated using Monte Carlo simulation, why was an OLS model used? What advantage does this offer over the Bayesian spatial model which has already been performed? (The explanation on p12 lines 360 to 362 isn’t convincing. A Bayesian model could just as easily do this).

Reviewer #2: -Are the objectives of the study clearly articulated with a clear testable hypothesis stated? No

-Is the study design appropriate to address the stated objectives? Yes

-Is the population clearly described and appropriate for the hypothesis being tested? Yes

-Is the sample size sufficient to ensure adequate power to address the hypothesis being tested? Yes

-Were correct statistical analysis used to support conclusions? Yes

-Are there concerns about ethical or regulatory requirements being met? Yes

**Results**

-Does the analysis presented match the analysis plan?

-Are the results clearly and completely presented?

-Are the figures (Tables, Images) of sufficient quality for clarity?

Reviewer #1: Figure 4: It would be useful having a map inset for the urban area of Caratinga in each plot (similar to Figure 7) to improve clarity.

Reviewer #2: -Does the analysis presented match the analysis plan? Yes

-Are the results clearly and completely presented? Yes

-Are the figures (Tables, Images) of sufficient quality for clarity? No

**Conclusions**

-Are the conclusions supported by the data presented?

-Are the limitations of analysis clearly described?

-Do the authors discuss how these data can be helpful to advance our understanding of the topic under study?

-Is public health relevance addressed?

Reviewer #1: Page 10 to 14: The Discussion should be limited to a discussion of the results of the study, limitations of the methods, caveats, and suggestions for future research. There is a lot of content here that is a reiteration of the introduction or new background (e.g. lines 271-280, lines 377-395, lines 397-405), introduction of new data/methods (e.g. lines 305), results rather than a discussion of them (e.g. lines 286-293, lines 395-396), and otherwise superfluous content like explanation of methods (e.g. lines 308, 326). Additionally, the purpose of the two hypotheses mentioned on line 293 is unclear. Is this speculation or a suggestion for future research to address?

Reviewer #2: - Are the conclusions supported by the data presented? Yes, but need to get better

-Are the limitations of analysis clearly described? Yes

-Do the authors discuss how these data can be helpful to advance our understanding of the topic under study? Yes

-Is public health relevance addressed? Yes

**Editorial and Data Presentation Modifications?**

Reviewer #1: p4, lines 120 to 122: I suggest a rewording of this sentence to be clearer, e.g. “Depending on the city, peridomiciliary/urban transmission and/or sylvatic transmission may be observed.”

p5, lines 143 to 151: I don’t know why the study area and statement on ethics approval have been lumped together. I think ethics approval would be better suited elsewhere.

p5, line 151: This last sentence is unclear. Does this mean all data involved was de-identified? This seems to be stated on line 155 already, so I suggest omitting it here.

p5, lines 157 to 158: Why was it necessary to categorise patients into three-year intervals to analyse their spatial distribution? Is it because there are too few cases for each year individually, or some other reason? It would be helpful to add this reasoning.

p5, lines 160 to 162: The terms “…empirical spatial smoothing” and Bayesian “spatial smoothing” don’t need to be capitalised.

p5, lines 161 to 162: I may be misunderstanding the term “accumulated” in this sentence, but the incidence rate should be estimated for each combination of area and demographic sector. However, only the demographic sectors are explicitly mentioned.

p5, line 163: The term “queen” is out of place in the sentence. Moreover, it is not a very useful term when relating to non-lattice geographic units. I suggest describing the spatial weights matrix as “A first-order adjacency matrix”, as is common practice with this type of spatial data.

p6, line 168: The first sentence here isn’t clear (the word “ellipse” by itself does not indicate what type of analysis/test is being performed). I suggest deleting this sentence, and begin this section with “Directional distribution ellipses provide…”.

p6, line 174: Similarly, the word “kernel” is not very descriptive. I suggest using “kernel density maps”.

p6, line 190 to 191: The reader should be familiar with the assumptions of OLS and it is not necessary to state this.

p6, lines 191 to 193: I suggest rephrasing this sentence to make it clearer these items were observed rather than estimated: “Population, garbage dumped…, sewage collection were included as explanatory variables”.

p6, line 194: The meaning of “according to the variables” is unclear. Does this mean AIC is computed for model including all explanatory models?

p7, line 202: The abbreviation “HP” is unclear.

p7, line 215: Use of parenthesis is inconsistent.

p13, line 373: “analyzes” should be “analyses”. Similarly on p14, line 411.

p13, line 373: “this” should be “which”.

p14, line 413: Suggest rewording the beginning of this sentence to “The spatial analysis was carried out…”.

Figure 2: I suggest adding the word “district” in parenthesis to the second legend item, Patrocinio de Caratinga.

Reviewer #2: Title - include the title search for sand fly.

Abstract - I suggest reducing the introduction of the abstract, which must be extremely brief, and place the reader on the subject covered in the study. Add the objectives and improve the description of the results, adding relevant discussions.

Pag. 1, Line 30 - "Information was retrieved from the public health archives", all information? Specify which.

Pag. 1, Line 40 - Improve the conclusion of the summary. What does the study bring again?

"Based on our analyzes, the recent urbanization of ATL in Caratinga…" this information is already on the title.

Introduction - Include sand fly research and leishmania research in the objectives

Methodology

Pag 9, line 156 - add education level

Study area - briefly characterize the municipality studied. PIB, climate, main economic production, so that whoever lives in another state or country, can understand the characteristic of the study area.

Results

Pag. 7, line 214 - 319 cases notified in which period? Please specify.

Pag. 8, line 242 - I suggest removing "Confirming our previous observations", this statement is unnecessary in this paragraph.

Pag. 8, line 245 - add this information to the methodology and describe only the result.

Pag. 8, line 250 - add this information to the methodology and describe only the result.

Discussion

Pag. 10, line 300 – which places? It will be interesting to know some of them. 

Pag. 14, line 428 - The conclusion must be aligned with the objectives.

Figure 4 - specify in the map legend and the figure legend if it is about the incidence and per 10,000 inhabitants. District names are unreadable in some places. I suggest removing it since the map specifying the districts with the names has already been presented previously.

**Summary and General Comments**

Reviewer #1: (No Response)

Reviewer #2: The spatial analysis of cases of ATL brings relevant information about the distribution of the disease and the spatial statistics to show the dependence of the space in the occurrence of the disease and allows the knowledge of the associated factors. These are important analyzes for a better understanding of the epidemiology of the disease and generate information that assists in the decision-making of health agencies for the prevention of the disease. The authors were careful to analyze environmental factors, such as the presence of trees, and this enriched the study. I made some observations to improve the quality of the manuscript.

PLOS authors have the option to publish the peer review history of their article (what does this mean?). If published, this will include your full peer review and any attached files.

Reviewer #1: Yes: Earl W. Duncan

Reviewer #2: Yes: Melca Niceia Altoé de Marchi
---

## [Decision Letter · Decision Letter 1]

30 Apr 2021

Dear Dr Soares,

We are pleased to inform you that your manuscript 'A follow-up study (2007-2018) on American Tegumentary Leishmaniasis in the municipality of Caratinga, Minas Gerais State, Brazil: spatial analyses and sand fly collection' has been provisionally accepted for publication in PLOS Neglected Tropical Diseases.

Best regards,

Shan Lv, Ph.D.

Deputy Editor

Shan Lv

Deputy Editor

Reviewer's Responses to Questions

**Key Review Criteria Required for Acceptance?**

**Methods**

-Are the objectives of the study clearly articulated with a clear testable hypothesis stated?

-Is the study design appropriate to address the stated objectives?

-Is the population clearly described and appropriate for the hypothesis being tested?

-Is the sample size sufficient to ensure adequate power to address the hypothesis being tested?

-Were correct statistical analysis used to support conclusions?

-Are there concerns about ethical or regulatory requirements being met?

Reviewer #1: (No Response)

Reviewer #2: -Are the objectives of the study clearly articulated with a clear testable hypothesis stated? Yes

-Is the study design appropriate to address the stated objectives? Yes

-Is the population clearly described and appropriate for the hypothesis being tested? Yes

-Is the sample size sufficient to ensure adequate power to address the hypothesis being tested? Yes

-Were correct statistical analysis used to support conclusions? Yes

-Are there concerns about ethical or regulatory requirements being met? Yes

**Results**

-Does the analysis presented match the analysis plan?

-Are the results clearly and completely presented?

-Are the figures (Tables, Images) of sufficient quality for clarity?

Reviewer #1: (No Response)

Reviewer #2: -Does the analysis presented match the analysis plan? Yes

-Are the results clearly and completely presented? Yes

-Are the figures (Tables, Images) of sufficient quality for clarity? Yes

**Conclusions**

-Are the conclusions supported by the data presented?

-Are the limitations of analysis clearly described?

-Do the authors discuss how these data can be helpful to advance our understanding of the topic under study?

-Is public health relevance addressed?

Reviewer #1: (No Response)

Reviewer #2: -Are the conclusions supported by the data presented? Yes

-Are the limitations of analysis clearly described? Yes

-Do the authors discuss how these data can be helpful to advance our understanding of the topic under study? Yes

-Is public health relevance addressed? Yes

**Editorial and Data Presentation Modifications?**

Reviewer #1: p8, line 240: “incindente” should be incidence.

p8, line 242: “Directional Distribution Ellipses” doesn’t need to be capitalized.

Reviewer #2: All my concerns was attended. I recommend this manuscript to be publish.

**Summary and General Comments**

Reviewer #1: (No Response)

Reviewer #2: All my concerns was attended. I recommend this manuscript to be publish.

PLOS authors have the option to publish the peer review history of their article (what does this mean?). If published, this will include your full peer review and any attached files.

Reviewer #1: **Yes: **Earl W. Duncan

Reviewer #2: **Yes: **Melca Niceia Altoé de Marchi

---

## [Editor Report · Acceptance letter]

13 May 2021

Dear Mr. Soares,

We are delighted to inform you that your manuscript, "A follow-up study (2007-2018) on American Tegumentary Leishmaniasis in the municipality of Caratinga, Minas Gerais State, Brazil: spatial analyses and sand fly collection," has been formally accepted for publication in PLOS Neglected Tropical Diseases.

Best regards,

Shaden Kamhawi

co-Editor-in-Chief

Paul Brindley

co-Editor-in-Chief
